# Stable laser-acceleration of high-flux proton beams with plasma collimation

M. J. V. Streeter [1], G. D. Glenn [2,3], S. DiIorio [4], F. Treffert [2,5,6], B. Loughran [1], H. Ahmed [7], S. Astbury [7], M. Borghesi [1], N. Bourgeois[7], C. B. Curry [2,8], S. J. D. Dann [7], N. P. Dover [9], T. Dzelzainis [7], O. C. Ettlinger[9], M. Gauthier [2], L. Giuffrida [10], S. H. Glenzer [2], R. J. Gray[11,12], J. S. Green[7], G. S. Hicks[9], C. Hyland [1], V. Istokskaia [10,13], M. King [11,12], D. Margarone[1,10], O. McCusker[1], P. McKenna [11,12], Z. Najmudin [9], C. Parisuaña [2,14], P. Parsons[1], C. Spindloe[7], D. R. Symes[7], A. G. R. Thomas [4], N. Xu[9] & C. A. J. Palmer [1] ✉

Laser-plasma acceleration of protons offers a compact, ultra-fast alternative to conventional acceleration techniques, and is being widely pursued for potential applications in medicine, industry and fundamental science. Creating a stable, collimated beam of protons at high repetition rates presents a key challenge. Here, we demonstrate the generation of multi-MeV proton beams from a fast-replenishing ambient-temperature liquid sheet. The beam has an unprecedentedly low divergence of 1° (≤20 mrad), resulting from magnetic self-guiding of the proton beam during propagation through a low density vapour. The proton beams, generated at a repetition rate of 5 Hz using only 190 mJ of laser energy, exhibit a hundred-fold increase in flux compared to beams from a solid target. Coupled with the high shot-to-shot stability of this source, this represents a crucial step towards applications.

Ultra-short energetic proton beams are routinely produced in the interaction of high-intensity lasers with thin, solid targets[1–3]. The attractive properties of the generated proton beams, such as high-peak current (kA), low emittance (μm · mrad) and short bunch duration (≤ps at source) make them exciting potential sources for a large number of applications[4], including fundamental physics[5,6], inertial confinement fusion[7] and materials science[8]. There is also a concerted effort to utilise laser-driven proton beams for radiobiology and particle therapy studies[9–13]. This is motivated by proof-of-concept experiments indicating the beneficial effects of exposure to radiation at high-dose rates (FLASH effect at > 40 Gy·s⁻¹) with reduced toxicity to healthy tissues[14]. The underlying mechanisms of this effect are still subject to debate[15].

For relativistic laser interactions with micron-thick targets, proton acceleration occurs in a TV · m⁻¹ electrostatic sheath field formed by the charge separation between laser-heated electrons and the target surface[16,17]. When using planar targets, the acceleration direction is typically perpendicular to the target plane, and so is commonly referred to as target-normal sheath acceleration (TNSA). Despite their unique beam properties, exploitation of laser-driven proton accelerators for key applications, or as injectors into conventional beam transport lines, has been obstructed by the inherently large beam

¹School of Mathematics and Physics, Queen's University Belfast, Belfast, UK. ²SLAC National Accelerator Laboratory, Menlo Park, CA, USA. ³Department of Applied Physics, Stanford University, Stanford, CA, USA. ⁴Gérard Mourou Center for Ultrafast Optical Science, University of Michigan, Ann Arbor, MI, USA. ⁵Institut für Kernphysik, Technische Universität Darmstadt, Darmstadt, Germany. ⁶Lawrence Livermore National Laboratory, Livermore, CA, USA. ⁷Central Laser Facility, STFC Rutherford Appleton Laboratory, Didcot, UK. ⁸Department of Electrical and Computer Engineering, University of Alberta, Edmonton, AB, Canada. ⁹The John Adams Institute for Accelerator Science, Imperial College London, London, UK. ¹⁰ELI Beamlines Facility, The Extreme Light Infrastructure ERIC, Dolní Břežany, Czech Republic. ¹¹Department of Physics, SUPA, University of Strathclyde, Glasgow, UK. ¹²The Cockcroft Institute, Sci-Tech Daresbury, Warrington, UK. ¹³Faculty of Nuclear Sciences and Physical Engineering, Czech Technical University in Prague, Prague, Czech Republic. ¹⁴Department of Mechanical Engineering, Stanford University, Stanford, CA, USA. ✉e-mail: c.palmer@qub.ac.uk

divergence (≥100 mrad[18,19]) which causes rapid increase in beam size and reduction of particle flux with distance from the source.

While beam-capturing systems are being explored[10,13,20–22], a significant reduction in the initial beam divergence would greatly improve the utility of these schemes and mitigate the need for capturing in some cases. Previously, shaping of the target surface has been used to modify the morphology of the sheath field and create focusing effects over short (<1 mm) distances[23,24]. More sophisticated targetry has also been employed to create a focusing and accelerating electromagnetic pulse in a helical coil attached to the target[25]. Of these, the latter has demonstrated the highest suitability for coupling of protons into additional beam transport systems, but the complexity of the target presents a challenge for high repetition rate (>1 Hz) operation.

In this article, we report on the experimental demonstration of low divergence (≤20 mrad rms) proton beams generated by the interaction of a multi-TW laser pulse with a water sheet target. Approximately 0.5% of the 190 mJ laser pulse energy was converted to the proton beam, which had a maximum detected energy of 6 MeV. The plasma accelerator exhibited high stability relative to typical laser-plasma ion sources, with beam properties varying on the 10% level. This resulted in a compact proton source capable of reproducibly delivering ≥40 Gy to a diagnostic placed 160 mm from the interaction at a repetition rate of 5 Hz. Higher proton flux and peak energy have been previously observed from sub-Hz repetition rate experiments with significantly higher laser power[26,27]. However, the results presented here are a significant advance through the simultaneous realisation of a stable multi-Hz laser-proton accelerator with an order of magnitude reduction in proton beam divergence and a hundred-fold increase in flux in comparison to using solid (Kapton tape) targets[28] with the same experimental system.

## Results

### Generation of low divergence proton beams

We focused ultra-high intensity laser pulses onto the surface of a continuously-flowing ambient-temperature liquid water jet target[29], which can provide a variable thickness (0.2–5 μm) water sheet with a kHz-compatible refresh rate. Previous experiments have shown kHz-operation of a similar liquid target to produce 2 MeV proton beams but with comparatively low flux (divergence was not measured)[30]. Here, the water sheet surface normal was aligned at 30° to the laser axis, as illustrated in Fig. 1a, and the target thickness was (600 ± 100) nm at the interaction point, where the error represents the calibration uncertainty as opposed to fluctuations in the thickness. The sheet was monitored and remained stable (with position fluctuations < 5 μm) over the duration of the measurements.

Electrons in the target were heated by the laser, with the escaping electron energy distribution sampled by an electron spectrometer in the laser forward direction. A proton beam was observed along the direction of the target rear-surface normal (unirradiated side), as measured by the proton beam profiler 160 mm behind the target. Compared to irradiation of a reference 12.7 μm-thick Kapton tape target[28], the divergence and peak dose of proton beams from the water target were substantially altered. For the tape target, large divergence (>100 mrad) beams were observed, typical for sheath acceleration, with a peak dose of 0.5 Gy (Fig. 1b). By contrast, the proton beams from the water target had an order of magnitude lower divergence, as low as (12 × 20) mrad$^2$, and the peak dose generated by individual laser pulses increased by two orders of magnitude to 55 Gy (Fig. 1c) for a Bragg peak proton energy of 1.1 MeV. While some decrease in beam divergence can be attributed to the reduced target thickness, this is not sufficient to explain our results[31].

The proton spectrum was simultaneously sampled by a time-of-flight (TOF) spectrometer, placed 3° from the nominal target rear-surface normal direction. The resulting relative spectra were combined with the peak dose observed on the proton spatial profile to infer the absolute spectral intensity on the beam axis. Comparison of the proton spectra for the tape and water targets (Fig. 1d) show that the spectral intensity increased by two orders of magnitude for the water target (to ~10$^{11}$ protons MeV$^{-1}$·sr$^{-1}$, for (1 ≤ $E_p$ ≤ 4) MeV). The total observed charge of protons above 1 MeV also greatly increased from (8$^{+12}_{-8}$) pC to (270 ± 60) pC (average and rms). This was accompanied by an increase in the maximum detectable proton energy from 4 MeV to 6 MeV and a significant improvement in relative shot-to-shot stability of the spectrum (flux fluctuations decreasing from 174% to 46% (rms) over the range 1–3 MeV).

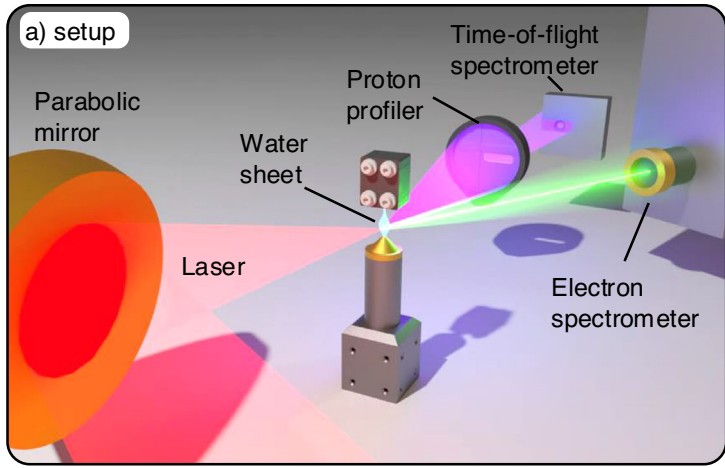

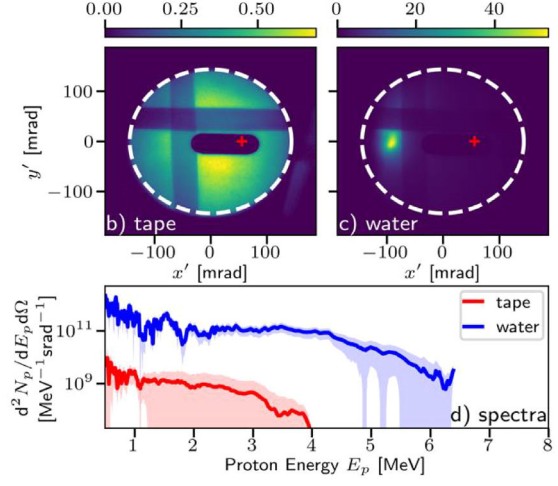

**Fig. 1 | Experimental setup and generated proton beams. a** Illustration of the experimental setup, showing the laser-water interaction and the primary diagnostics. Each laser pulse contained up to 200 mJ in a pulse length of (57 ± 5)fs (FWHM) and was focused to a focal spot waist of (1.2 × 1.4) μm$^2$, giving a peak intensity of $I_0$ = (3.5 ± 0.4) × 10$^{19}$ W cm$^{-2}$. The laser was focused onto a thin ((600 ± 100) nm) sheet of water generated by a converging nozzle geometry in a vacuum chamber. **b** Example proton dose distributions for the beams produced from single shots with the 12.7 μm thick Kapton tape target and **c** the water sheet using comparable laser settings and the same detector screen. The dashed white line highlights the edge of the scintillator screen and the horizontal and vertical stripes are due to aluminium filters which blocked lower energy protons. The dark oval was a hole in the scintillator screen permitting a line-of-sight for the time-of-flight (TOF) diode, as indicated by the red '+'. **d** The average proton spectra, recorded by the TOF diagnostic, were calculated from several shots at the same conditions as (**b**) (red−average of 20 shots) and (**c**) (blue−average of 50 shots). The spectra are normalised to peak flux observed on the scintillator screen. The solid lines show the average spectrum, while the shaded region shows the rms shot-to-shot variations.

The proton beam axis could be controlled by translating the 350 μm wide water sheet in the horizontal plane, indicating a radius of curvature of the rear surface of $R_x \approx 1.3$ mm. The shape and stability of the spectrum was relatively constant while changing the angle of the proton beam from $x' = -65$ mrad to $x' = 52$ mrad (relative to the nominal rear surface normal), indicating that the spectrum measured by the TOF was relatively independent of proton beam pointing. The curvature in the vertical plane was determined to be much larger ($R_y \gtrsim 100$ mm), and so the low divergence of the proton beam cannot be explained by target curvature. Further information on the beam steering is given in the Supplementary Information.

The spatial profile measurements over the minimally filtered regions of the profile screen (labelled region 1 in Fig. 2) are dominated by 1.1 MeV protons. Additional aluminium filters were placed over some regions of the screen increasing the dominant proton energy, thereby allowing measurement of the spatial profile for different energy bands. Figure 2a shows the spatial dose distribution of a proton beam which was steered to the crossing point of the filters. The reduction in signal in each differently filtered region was consistent with the proton spectrum independently measured by the TOF spectrometer. Under the assumption that the proton spectrum was spatially independent, the proton flux (for protons with $E_p > 1$ MeV) was calculated for each part of the screen, as shown in Fig. 2b. The self-consistent beam shape across the different filter regions indicates that the proton beam had a low divergence over the 1.1–3.6 MeV energy range.

## Stable high quality proton beam acceleration

The high stability of the proton beam generated by the laser-water interaction is qualitatively shown in the 10 consecutive shots presented in Fig. 3a which demonstrates consistent proton beam spatial profiles, peak dose values and pointing. The beam properties are quantified in Fig. 3b, c which show the beam divergence (from 2D Gaussian fits) and centroid position values for 300 consecutive shots. The average and standard deviation beam divergences are (24 ± 2) mrad and (40 ± 3) mrad in the minor and major axes of the fitted ellipses respectively, showing both low average divergence and remarkable reproducibility. The centroids exhibit gradual drifts of ~10 mrad with random shot-to-shot fluctuations of ≤ 5 mrad. For comparison, a burst of 40 shots on the reference Kapton tape produced proton beams with a much larger average divergence of (120 ± 30) mrad and pointing fluctuations of 9 mrad.

The peak dose observed on the proton beam profiler was also stable, exemplified over the same 300 shots on the water sheet target (plotted in Fig. 3d) with an average and standard deviation of $D = (41\pm5)$ Gy deposited by protons with an average energy of 2.1 MeV. Similarly, the proton spectral shape was also reproducible shot-to-shot with a local maximum at (4.1±0.1) MeV, as shown in Fig. 3e. The relative shot-to-shot variation in the peak dose was approximately equal to the relative variation in the focused laser intensity.

## Proton energy scaling with laser energy and focus

Figure 4 shows the dependence of electron and proton beam properties on the laser focus and pulse energy. Figure 4a–c show the results of translating the target plane along the laser propagation axis while maintaining a constant focal plane of the laser pulse, thereby modifying the maximum intensity incident on the target surface. Zero defocus ($z_T = 0$), otherwise termed best focus, was taken as the position for which the maximum electron charge ($N_e$) was detected, which was also the centre of symmetry, i.e $N_e(z_T) \approx N_e(-z_T)$, as shown in Fig. 4b. The electron temperature, $T_e$, (Fig. 4c) was slightly higher at 0.71 MeV for $z_T = 22$ μm compared to 0.63 MeV at $z_T = 0$, where positive values of $z_T$ correspond to the laser focusing before the target surface. This may imply that the laser intensity, $I_L$, was slightly higher at that position ($T_e \sim [I_L\lambda_L^2]^{0.5}$, where $\lambda_L$ is the laser wavelength[16]).

We typically observed non-Maxwellian proton spectra with no clear cut-off energy. To provide a useful measure of the 'maximum' proton energy we quote the 95% percentile energy, which was maximised at (4.4 ± 0.1) MeV for $z_T = 0$, as shown in Fig. 4a. Both proton

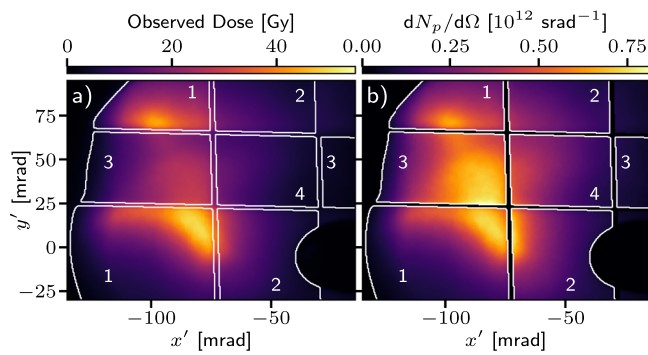

**Fig. 2 | Energy dependant proton beam spatial profiles. a** The observed dose profile for a single shot. The different filtered regions are labelled as 1) aluminised mylar filter plus 2–4) 10 μm, 20 μm and 30 μm of additional aluminium respectively. The peak energy deposition (Bragg peak) occurs for 1.1 MeV, 1.5 MeV, 1.9 MeV and 2.2 MeV protons in regions 1-4 respectively. The average energy of protons contributing to the signal in each region (weighted by the measured proton spectrum and relative dose deposition per particle) is 2.2 MeV, 2.8 MeV, 3.3 MeV and 3.6 MeV. The peak dose for each region was 50 Gy, 36 Gy, 32 Gy and 20 Gy respectively. **b** The flux of protons with $E_p \geq 1$ MeV inferred using the relative spectrum provided by the TOF spectrometer.

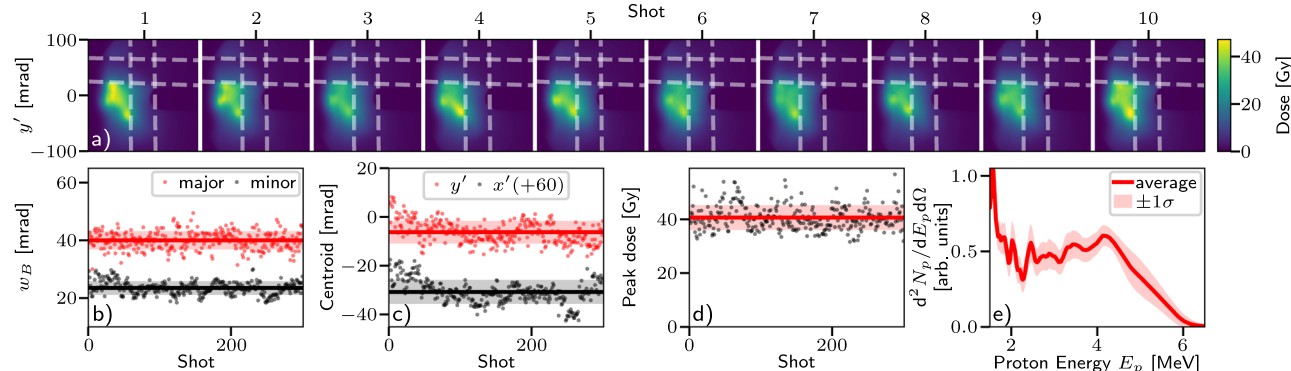

**Fig. 3 | Proton beam stability at 5Hz operation. a** Measured dose profiles from 10 consecutive shots with the water sheet target and nominally identical conditions. The horizontal and vertical bands are created by aluminium filters as indicated by the dashed white lines. **b, c** The rms beam waists and centroids of 2D Gaussian fits to the unfiltered region of the proton spatial profile and (**d**) the peak dose observed for 300 consecutive shots. **e** The average proton spectrum as recorded by the TOF spectrometer. For (**b–e**) the standard deviation is indicated by the shaded region.

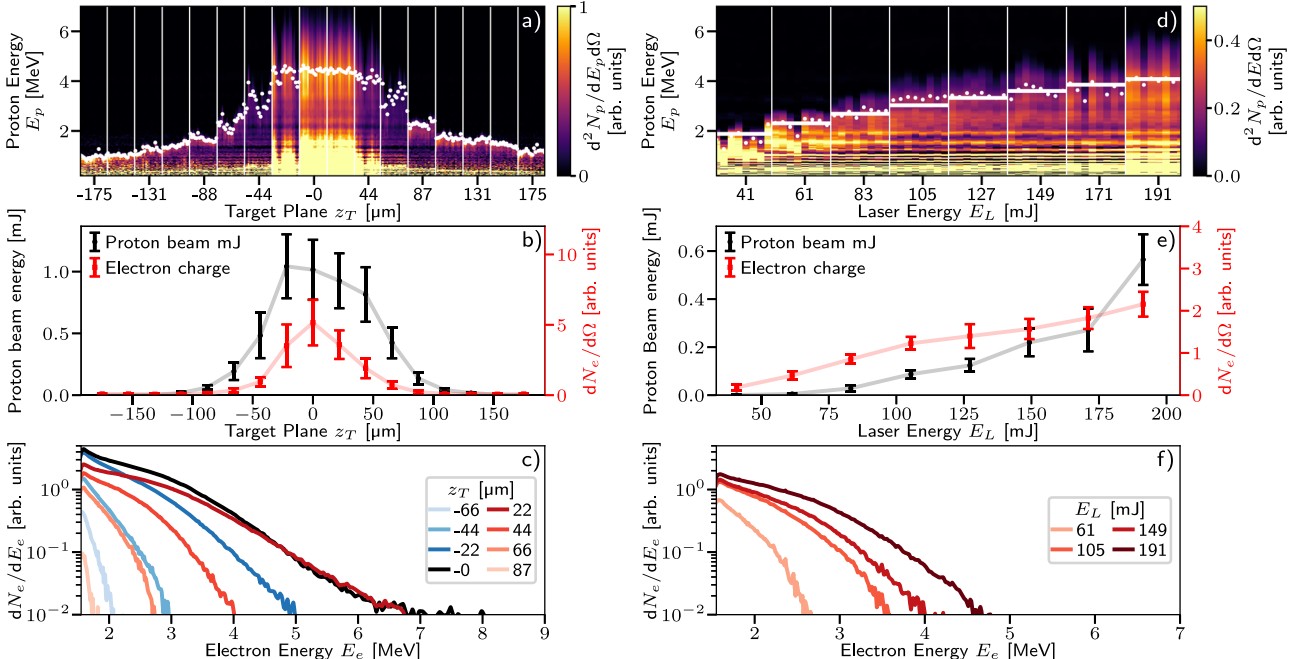

**Fig. 4 | Proton and electron beam parameter scans. a, d** Waterfall plots showing individual proton spectra for automated scans of target position along the laser propagation axis $z_T$ and laser energy $E_L$ respectively. The spectra are taken in bursts of equal $z_T$ and $E_L$, as indicated by the vertical dividing lines. The horizontal bands visible in the plots are due to spurious electrical noise in the diagnostic. The 95% percentile proton energies for each shot are overlaid as white dots. **d** also shows the predicted maximum proton energies from the best fitting $E_p \propto E_L^{1/2}$ scaling as horizontal white lines. **b, e** The total detected proton beam energy (left axis) and the detected electron charge (right axis) shown as the average and standard deviation of each burst. **c, f** The average electron spectra for shots at selected values of $z_T$ and $E_L$. Positive values of $z_T$ correspond to the laser focusing before the target surface, while negative target positions correspond to the laser interacting with the target before reaching best focus. Note that the energy scan shown in (**d–f**) was taken for a slight defocus of $z_T \approx 30\,\mu m$.

energy and flux, the latter of which is shown in Fig. 4b, were maximised for best focus, but the proton flux dropped relatively gradually with defocus compared to the electron beam flux. The relative insensitivity of the proton beam acceleration to defocus is partly responsible for the high level of proton beam stability seen in Fig. 3. Moving the target from $z_T = 0$ to $z_T = 22\,\mu m$, a distance slightly greater than the Rayleigh range of the focusing optic ($Z_R \approx 15\,\mu m$), did not significantly change the maximum proton energy (($4.44 \pm 0.09$) MeV and ($4.39 \pm 0.07$) MeV respectively).

The scaling of the proton and electron beam properties with laser energy can be seen in Fig. 4d–f. The proton and electron maximum energies and flux increase with laser energy, with the proton maximum energy consistent with a $E_p[\text{MeV}] = 9(E_L[\text{J}])^{1/2}$ scaling as indicated by the horizontal lines in panel Fig. 4d. However, the total proton beam energy was seen to scale approximately linearly with laser energy, consistent with a relatively constant conversion efficiency, indicating promising performance with commonly available higher energy laser systems[32]. If the energy scaling and collimating effect can be maintained at higher energies, then proton beams with energies of 40–90 MeV and sub-degree divergence may be achievable with modern 10–100 J laser systems.

The high peak dose obtained here with 190 mJ indicates that, with the addition of suitable proton beam transport, this source is suitable for high dose-rate radiobiology[12]. The source can readily be scaled to higher repetition rates, with kHz laser systems now entering this energy regime[32], thereby providing a high flux kHz source of 4 MeV protons for applications in materials science[8,33].

## Discussion

Numerous preceding studies of TNSA have consistently observed divergences of ≥100 mrad, over a wide range of laser and target parameters[2,31,34–36]. Lower divergence proton beams have been reported from defocused laser interactions with ultra-thin (≤10 nm) foils[37],

but the stringent target requirements and the need to pre-condition the laser with plasma mirrors make even moderate repetition rate (~1 Hz) operation a considerable technological challenge. Although other laser-driven proton acceleration mechanisms have produced lower divergences, for example in collisionless shock acceleration[38], or in ablation plasmas[39], these schemes suffer from significantly reduced flux and/or particle energy.

The inherently high-divergence of TNSA proton beams has been attributed to curvature of the plasma surface, which develops as the plasma expands during the acceleration process[40]. During the early stages of propagation, the protons are assumed to be accompanied by a population of electrons that act to mitigate space-charge effects allowing the proton propagation to be modelled as ballistic with no further lateral acceleration[41]. The observed laminarity of TNSA proton beams implies the lateral distribution of co-propagating electrons is well described by electron temperatures on the order of 100 eV[42].

In our experimental setup, the presence of low-pressure water vapour resulted in the stable generation of low divergence high-flux proton beams by supporting the generation of focusing magnetic fields around the ballistically propagating beam of protons and co-propagating electrons. The background vapour density is sufficiently low so that it does not significantly impact the initial acceleration process[43], but does affect the subsequent propagation of the proton beam. This is apparent in numerical simulations of a charge-neutralised proton beam through an initially neutral background of water vapour performed using the particle-in-cell code OSIRIS[44,45]. While these simulations necessarily represent a simplified study of proton propagation through a vapour due to the high computational cost of modelling the multi-scale physics involved, they provide insight into the key physical processes driving the reduction in beam divergence (full detail of the simulation setup is included within the Methods).

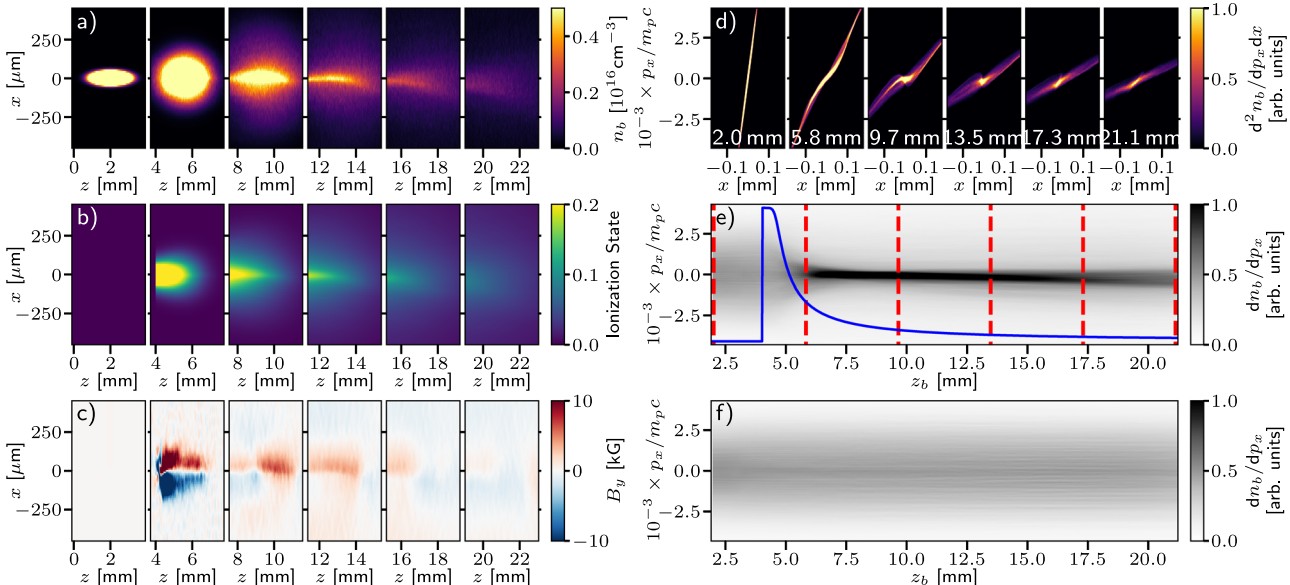

**Fig. 5 | Proton focusing in particle-in-cell simulations.** Series of snapshots of (**a**) the proton beam density $n_b$, **b** the water molecule ionisation state, **c** the azimuthal magnetic fields and **d** the proton beam transverse phase space as the proton beam propagates through the vapour. The progress of the proton beam in each snapshot is indicated by the $z$-axis (**a**–**c**) and the text labels (**d**) (note that the bunch stretches longitudinally due to velocity dispersion). **e, f** Proton beam angular distributions as functions of the proton beam centroid position $z_b$ for propagation through vapour and vacuum respectively. Both plots are normalised to the same value to allow direct comparison. The red dashed vertical lines in (**e**) indicate the corresponding positions of the snapshots in (**a**–**d**) and the blue line indicates the normalised water vapour density profile (scaled to the plot window), which had a maximum molecular density of $6 \times 10^{17}$ cm$^{-3}$.

The proton beam was initialised with a rms tranverse and longitudinal beam size of 20 μm and 500 μm respectively. The initial divergence was 20 mrad with a transverse phase-space correlation corresponding to a normalised emittance of 2 μm mrad. The normalised proton beam momentum was $p_z/m_e c = 0.1$, with a rms spread of 20%. The electrons were initialised with the same spatial distribution and forward drift velocity as the proton beam but with a thermal uncorrelated momentum spread corresponding to a temperature of 200 eV transverse to the propagation direction. This temperature is comparable with estimates of proton beam temperature inferred from measurements of beam emittance[42] and is sufficiently low to prevent a space-charge driven increase in proton beam transverse momentum during vacuum propagation. The beams were initialised in vacuum before propagating into the water vapour with a density profile determined by an empirical fit to fluid simulations of the water vapour profile (details in Methods).

As the beam propagates (Fig. 5a), the water vapour is readily ionised (primarily by the protons[46]) producing a population of cold electrons ($T_{ec}$ ~ 10–100 eV). The vapour is ionised to produce up to 0.8 electrons per water molecule over the area occupied by the beam (Fig. 5b). The proton-electron beam propagating through this cold plasma is then subject to the ion filamentation instability[47,48] causing rapid growth of an azimuthal magnetic field, which is focusing for the proton beam[49].

Due to the low density of the background plasma and high relative velocity of the energetic protons, their interaction is dominated by collisionless mechanisms. The transverse diffusion rate of the proton beam particles, due to collisions between the fast protons and the stationary ions, is $\nu_\perp^{i|i} \lesssim 100$ s$^{-1}$, for a proton beam velocity of $v_D$ ~ 0.1c and $n_{i,\mathrm{plasma}}$ ~ $10^{17}$ cm$^{-3}$ [50]. The proton beam intra-beam scattering frequency (~$10^8$ s$^{-1}$ for a beam temperature of 200 eV[51]) is also significantly below the growth rate for the ion Weibel instability[47,52] (~$v_D/\delta_i \approx 10^{10}$ s$^{-1}$). This allows the collisionless instability to develop with a maximal growth rate for length scales on the order of the ion skin depth of, $\delta_i = c/\omega_{pi}$ where $\omega_{pi} = \sqrt{(Ze)^2 n_i/m_i \varepsilon_0}$ is the ion plasma frequency. For the experimental conditions, this is on the order of 1 mm. This scale is sufficient to capture the whole proton beam within a single filament, with the pinching magnetic field saturating over the scale of several centimetres of propagation. Snapshots from the simulations illustrate the magnetic field which extends along the length of the proton bunch (Fig. 5c). The influence of this pinching azimuthal magnetic field is also apparent within the simulated proton phase space (Fig. 5d). These plots illustrate the initial linear correlation of a diverging laminar beam, which is then modified by the magnetic focusing. This focusing reduces the transverse momentum spread of the proton beam (reaching a minimum at $z_b \geq 10.0$ mm). From that point, the drop in vapour density means that the magnetic fields are much reduced and the beam slowly expands with a reduced divergence. The evolution of the proton beam angular distribution with and without the influence of the background vapour can be compared in Fig. 5e, f respectively, clearly highlighting the collimation of the proton beam within the background plasma over a long distance.

In reality, the proton beam distribution is more complex than our simplified simulations, however, the experiment shows the efficacy of harnessing this beam-plasma effect. Tailoring of vapour density profile presents an exciting opportunity to explore this mechanism for control of proton beam propagation over the metre scale for applications requiring high flux or high dose-rate MeV proton bunches.

In summary, our results demonstrate an approach for generating stable, low-divergence MeV proton beams from multi-Hz laser-plasma interactions, thereby overcoming several significant hurdles in the development of these plasma accelerators. In particular, the low divergence can enable vastly improved beam capture and transport, such as required for radiobiology experiments[13]. The stable high-repetition-rate operation would then allow studies in the FLASH regime (>40 Gy s$^{-1}$) using single or multiple exposures. Such a platform would be suitable for other applications requiring delivery of high flux proton radiation, such as materials damage testing[8] and fundamental physics of proton stopping in extreme states of matter[6,53].

The relatively modest laser energy requirements (190 mJ per pulse) and the high flow rate of the liquid target (10 m s$^{-1}$)[29] are compatible with kHz operation rates, and the interaction produces

significantly less debris compared to solid-foil targets. Therefore, it is possible to operate the source continuously at kHz for extended durations and to easily change the ion species produced by using different liquid targets, for example, the acceleration of deuterons from heavy water[54].

The simultaneous observation of several desirable beam properties at high repetition rate in this experiment is unprecedented in laser-driven proton acceleration. The collimating effect of a background low-density plasma has not been previously explored and opens up a new approach for the control and optimisation of laser-driven ion sources. Extending these results to still higher repetition rates and laser energies will provide a unique proton source for many important applications.

## Methods

### Laser setup
The experiment was performed with the Gemini TA2 Ti:sapphire laser system at the Central Laser Facility, using the arrangement shown in Fig. 1a. The energy of the laser was measured by a CCD camera, imaging the near-field of the laser through a dielectric mirror. This camera was cross-calibrated with a Gentec energy metre to provide on-shot measurements of the energy in each pulse. The spatial phase of the laser was measured and controlled using a HASO wavefront sensor and a piezo-electric adaptive optic. The spatial phase was flattened, to optimise the energy distribution in the final focal spot. The laser was focused with a 90° off-axis $f$/2.5 parabola, providing a peak intensity of $3.5 \times 10^{19}$ W cm$^{-2}$. The incidence angle of 30° relative to the target surface normal ensured a component of the laser electric field along the target normal (p-polarisation). A 0.5 mm-thick plate of fused silica was used to separate the target vacuum chamber from the compressor chamber. The compressed laser pulse passed through this plate, generating a maximum B-integral of 0.2. A 5 mm diameter sample of the compressed pulse was directed through an identical fused silica plate before being attenuated and passing through an additional 2 mm of fused silica onto a spectral phase diagnostic (LX Spider). The retrieved pulse was numerically back-propagated through 2 mm of fused silica to calculate the pulse at the interaction point.

### Water sheet target
The liquid sheet was created by the outflow of pressurised water from a tungsten nozzle[29,55]. The nozzle exit aperture was $(25 \times 100)$ μm$^2$, which shaped the flow of water to create wide sheets as the water was directed through the vacuum chamber. The sheet provided a variable thickness (0.2–5 μm) target with kHz-compatible refresh rate. The liquid sheet was characterised using white light interferometry, which provided a measurement of the thickness at the position of the high-intensity laser focus of (600±100) nm. The positional stability of the sheet was measured using optical probing perpendicular to the plane of the sheet. From this, the sheet edges were determined to have a positional jitter <5 μm. Based on the geometry of the sheet formation (e.g. the momentum of the single converging flow into the nozzle is transformed into momentum in the plane of the sheet), it is expected that jitter in the plane of the sheet will greatly exceed jitter perpendicular to the plane of the sheet (along the target normal direction). This indicates that the sheet surface remains well within the 15 μm Rayleigh range of the focusing laser.

In order to maintain vacuum, the liquid sheet was directed into a 'catcher' at the lower end of the sheet which was a heated unit with a small aperture that enabled the vapour to be efficiently evacuated as exhaust. While the water jet was running, the vacuum chamber maintained a pressure of 0.1 mbar.

Computational hydrodynamic simulations have been performed using OpenFoam [www.openfoam.org] to model the vapour density profile over the scale of 2 cm from the sheet. These 2D simulations approximate the sheet as a 500 μm wide rounded rectangle with a

250 μm radius of curvature. The surface of the sheet was set to a constant pressure 20 mbar inlet. The boundaries of the simulation box were modelled using wave-transmission boundary conditions with pressure fixed at 0.1 mbar at a distance of 0.5 m from the boundary to mimic the measured pressure at the vacuum gauge. The molecular number density profile of the vapour along the normal to the sheet surface was found to be well described by the following empirical fit,

$$n_m(z_m)[\text{m}^{-3}] = 0.6 \times 10^{24} \left( 1 + \left( \frac{z_m[\text{mm}]}{0.6} \right)^{6.9} \right)^{-0.16} \tag{1}$$

where $z_m$ is the distance from the sheet surface.

### Proton diagnostics
The spatial distribution of the proton beam was characterised using a 50 mm diameter ZnS(Ag) scintillator (EJ-440) which was placed 160 mm behind the target, centred on the target rear surface-normal. This proton profiler diagnostic was filtered with 12 μm of aluminium-flash-coated mylar over most of its area, making it most sensitive to 1.1 MeV protons, with additional aluminium vertical (10 μm) and horizontal strips (20 μm) used to provide regions with maximum sensitivity at 1.5 MeV and 1.9 MeV, respectively. The regions where the filters overlapped (30 μm) were most sensitive to 2.2 MeV. The energy deposition per proton as a function of proton energy was determined using FLUKA[56,57] simulations. The proton energy spectrum was measured using a 2 mm$^2$ diamond-diode in time-of-flight (TOF) configuration at 3° from the rear surface normal and 357 mm from the target surface.

The conversion of scintillator signal to deposited dose was done by cross-calibrating the measured scintillator emission with slotted radiochromic film (RCF) stack (a 2 piece stack of Gafchromic HDV2 and EBT3 behind a 12 μm thick aluminium-flash-coated Mylar shield) which was placed on the irradiated surface of the scintillator—the slots allowing samples of the beam to propagate unobstructed to the scintillator while the rest of the beam was captured by the RCF stack. With the tape target in place to produce a wide smooth beam, the scintillator signal was measured for 380 shots with the slotted RCF in position. The exposed HDV2 RCF was scanned to measure its optical density, which was then converted to dose using the calibration from[58] (using a known dose from a proton cyclotron). Comparing the deposited dose to the accumulated camera signal was then used to provide an absolute calibration. For shots with the water target, the large increase in dose saturated the CCD, and so a calibrated ND 0.6 filter was added and the camera gain was reduced. This combined to reduce the camera counts by a measured factor of 13.6, which was applied when calculating the dose profile for shots with the water target.

The stopping power of the proton beams in the scintillator was 2–4 orders of magnitude higher than electrons or photons respectively for the particle energy ranges generated by this experiment. In addition, the lower expected flux (as they are emitted over a large solid angle), means that the expected contribution of electrons and photons to the scintillator signal was negligible. The thin mylar foil between the source and the detector stopped heavier ions from reaching the detector.

For high stopping power radiation, quenching is a known issue with scintillation based detectors leading to a reduction in scintillation signal for a fixed quantity of deposited energy in comparison with signal resulting from exposure to radiation with lower stopping power. Quenching is typically estimated using the Birk's parameter, $kB$, which impacts the emitted scintillation light per unit of distance propagated,

d$L$/d$z$, via:

$$\frac{dL}{dz} = \frac{S \cdot (dE/dz)}{1 + kB \cdot (dE/dz)} \qquad (2)$$

where $S$ is the scintillation efficiency. Using this formula with the Birk's parameter for ZnS(Ag) of $kB = 2.4 \times 10^{-5}$ cm/MeV from[59] (for a scintillator density of 4.09 g/cm³) and maximum stopping power of the proton Bragg peak within ZnS(Ag) of $(dE/dz) \approx 1500$ MeV/cm, we obtain a negligible correction to the scintillation emission of $kB(dE/dz) \approx 0.04$.

### Electron spectral measurements

The spectrum of energetic electrons escaping the laser-plasma interaction was measured with a magnetic electron spectrometer placed behind the target in the laser forward direction (30° from the rear surface normal). A 2 mSr solid angle slit was placed at the entrance of the spectrometer, which used a compact (25 mm) 0.15 T dipole magnet to disperse electrons onto a Lanex scintillator screen. The scintillator screen was imaged onto a CCD camera. The energy dispersion of the spectrometer was determined using particle tracing through a 3D map of the magnetic field which was simulated using RADIA.

### Proton propagation simulation setup

The beam propagation through the low-density water vapour background was modelled in 2D Cartesian geometry using the particle-in-cell code OSIRIS[44,45], which includes a collisional ionization package using customisable cross-section data. The background water vapour was modelled as immobile neutral molecules with a number density given by equation (1) for $z_m = (z - 4)$ mm, where $z$ is the longitudinal axis of the simulation. The particle bunch was initialized in vacuum before propagating into the vapour to ensure that all protons traversed the complete density profile. The protons were allowed to interact with the water vapour solely via proton-impact ionisation. Water vapour ionisation cross sections due to proton-impact were calculated following[46]. Only the potential ionisation of the first ionisation state of the water molecule was considered, and up to 900 macro-particles per cell per allowed ionisation state were available for the freed electron population. No other ionisation or collisional effects were included. The ionised electrons were injected into the simulation and allowed to freely move, like the protons and electrons forming the beam, while the neutral vapour and vapour ions were modelled as a stationary background species. Identical simulations, except for the absence of the water vapour species, were used to compare the results to vacuum beam propagation.

The quasi-neutral proton and electron beams were initialised as Gaussian distributions with $n_b = n_0 \exp(-x^2/2w_x^2) \exp(-z^2/2w_z^2)$ with $n_0 = 1.1 \times 10^{17}$ cm⁻³, $w_z = 500$ μm (longitudinal) and $w_x = 20$ μm (transverse). The proton beam was initialised with a divergence of 20 mrad with a transverse phase-space correlation corresponding to a normalised emittance of 2 μm mrad. The normalised proton beam momentum was $p_z/m_p c = 0.1$, with a rms spread of 20%. The electron beam was given a thermal spread of 200 eV in all directions with a forward drift velocity of 0.1$c$. The proton and electron beams used 1296 and 36 macro-particles per cell respectively. The simulation domain was a 1300 × 240 cell grid spanning 26 mm in the $z$ direction and ±1.2 mm in the $x$ direction. Both simulations had a time-step of 29.8 fs.

## Data availability

The experimental data to support this study is freely available from the online repository Zenodo.org under the accession code https://doi.org/10.5281/zenodo.14236097.

## Code availability

The code used for analysis and presentation of the results is available in the data repository. The collisional ionisation module for OSIRIS is integrated into the OSIRIS code and available from the OSIRIS Collaboration.

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

## Acknowledgements

Special thanks goes to the staff at the Central Laser Facility who provided laser operational support, mechanical and electrical support, computational and administrative support. The authors would like to acknowledge the OSIRIS Consortium, consisting of UCLA and IST (Lisbon, Portugal) for providing access to the OSIRIS 4.0 framework. M.J.V.S. acknowledges support from the Royal Society URF-R1221874. S.H.G., G.D.G., C.P., M.G., C.C., F.T. acknowledge support from the U.S. DOE Office of Science, Fusion Energy Sciences under FWP No. 100182, and in part by the NSF Grant No. 1632708 and PHY-2308860. G.D.G. acknowledges support from the DOE NNSA SSGF programme under DE-NA0003960. A.G.R.T. and S.D. acknowledge support from the U.S. DOE Grant No. DE-SC0016804 and U.S. Air Force Office of Scientific Research Grant No. FA9550-19-1-0072 and U.S. Department of Energy NNSA Center of Excellence under cooperative agreement number DE-NA0004146. Z.N., O.E., G.H., and N.X. acknowledge support from the JAI, STFC grant no ST/P002021/1 and ST/V001639/1. P.McK, R.G and M.K. acknowledge support from EPSRC grant number EP/R006202/1. C.A.J.P. acknowledges support from EPSRC grant number EP/Y001737/1. B.L. acknowledges support from UK XFEL Physical Sciences Hub under agreement no. S2-2020-00020-8457. L. G. acknowledges support by the MŠMT ČR Project No. LQ1606 and by the project CZ.02.1.01/0.0/0.0/16_019/0000789.

## Author contributions

C.A.J.P. led conception and planning of the experiment with assistance from M.J.V.S., S.H.G., J.S.G., H.A., M.B., Z.N., R.J.G., N.P.D., P.McK.,

D.R.S., M.G., S.J.D.D., O.C.E., A.G.R.T., D.M., L.G., S.A., C.S., N.B., T.D. The target was developed at SLAC with input from G.D.G., F.T., C.B.C., M.G., and operated during the experiment by S.A., N.B., P.P., M.J.V.S., C.A.J.P. C.A.J.P. led the experimental construction and data acquisition, supported by M.J.V.S., B.L., G.S.H., O.C.E., N.X., D.R.S., C.H., O.McC., P.P., N.B., H.A., S.J.D.D., T.D. The diamond diode detector and control software were provided by D.M., L.G. M.J.V.S. led the analysis of the experimental data supported by C.A.J.P., B.L., G.D.G., V.I., F.T., C.P., M.G., N.P.D., O.McC. M.J.V.S. ran the computational fluid dynamics simulations of the liquid sheet. S.D. developed the collisional ionisation module for OSIRIS with support from A.G.R.T. S.D. and M.J.V.S. ran the OSIRIS simulations of proton beam propagation through the low-density background. M.B., N.P.D., M.G., J.S.G., S.H.G., R.J.G., D.M., B.L., V.I., L.G., Z.N., M.K., P.McK., P.P., C.S., A.G.R.T., S.D., D.R.S., F.T., G.D.G., M.J.V.S. and C.A.J.P. contributed to discussions of procedure, analysis and interpretation. F.T., A.G.R.T., D.R.S., P.P., Z.N., P.McK., S.H.G., D.M., B.L., R.J.G., G.D.G., N.P.D., S.D., M.B., M.G., J.S.G., V.I. contributed to writing of the manuscript co-ordinated by C.A.J.P. and M.J.V.S.

## Competing interests

The authors declare no competing interests.
