## [Transparent Peer Review file · Nature Communications]

Stable laser-acceleration of high-flux proton beams with plasma collimation

Corresponding Author: Dr Charlotte Palmer

Version 0:

Reviewer comments:

Reviewer #1

(Remarks to the Author)

The authors have made significant improvements to their manuscript and have responded to my comments and questions adequately, which I appreciate. I have a few remaining comments and questions regarding their line of argument about co-propagating electrons. They heavily rely on the Cowan 2004 paper to justify using ~ 200 eV co-propagating electrons in their explanation of the proton focusing effect and as input to simulation. I want to point out that, firstly, Cowan explicitly states that electrons are co-moving "during much of the acceleration", not during the drift phase (which is what the authors model in their OSIRIS simulation). In fact, Cowan shows in simulations that removing the co-propagating electrons after 1 cm doesn't affect the proton beam emittance, i.e., they don't contribute to the observed beam laminarity beyond 1 cm (well into the "drift phase"). Second, Cowan describes 15-200 eV as the "proton transverse temperature", rather than the electron temperature, which is what the authors are using these numbers for. They should justify why they assume the electron temperature is the same as the proton transverse temperature. Third, as the authors point out, electrons would have to have keV energies to keep up with the protons - does that mean that in their simulations, protons outrun electrons fairly quickly and there is actually only a brief co-propagation phase? In that case they may need to revise their explanation of the effect, where the co-propagating electrons play a minor role (because co-propagation is short).

I am happy to recommend this manuscript for publication in Nature Communications if the authors can address these remaining comments.

Reviewer #2

(Remarks to the Author)

This paper demonstrates the generation of multi-MeV proton beams with an unprecedentedly low divergence of 1σ (≤ 20 mrad) from a novel ambient-temperature liquid sheet with high shot-to-shot stability due to plasma collimation. The multi-Hz generation of laser-accelerated proton beams from laser to proton beam, it represents a step in advancing this source for future applications, as the proton flux or intensity is obviously enhanced by nearly 100 times compared to traditional TNSA regime.

However, it shows approximately 0.5% of the 190 mJ laser pulse energy was converted to proton beam, it means the acceleration mechanism is still in traditional TNSA regime. Therefore I still have some concerns as following:

As we saw in Physics of Plasmas 2013 Vol. 20 Issue 7, there were experimental studies of divergence of proton beams from nanometer thick diamond-like carbon foils irradiated by an intense laser with high contrast. Proton beams with also small divergence of 2 degrees are observed in addition with a well-collimated feature over the whole energy range, this also constitutes the smallest value nearly the same as in this paper, as a consequence, ~100 times increase in proton flux was also observed, while even with higher conversion efficiency from laser to proton beam.

Ionisation of the low-density vapour behind the target acts to limit the divergence growth of the proton beam, due to the protons propagation through the plasma where they are subject to magnetic pinching, this regime is maybe not applicable in higher laser intensity case and higher proton energy such as 60MeV, there are not any comments and extra simulations.

The paper demonstrates proton flux enhancement by plasma magnetic confinement of MeV proton beam, instead of new acceleration mechanism, which can also be realized by conventional magnets or capillary. I would recommend publication in other more technical journal.

Reviewer #3

(Remarks to the Author)

The novel version of the manuscript by M. Streeter et al., considerably gained in clarity, both on the underlining physics and on the scope of the paper.

The focusing mechanism that is responsible, according to the authors' evidence and analysis, for the collimation of the proton beam certainly deserves to be disclosed to the laser-plasma accelerator community, in order to stimulate further exploration. As far as I'm concerned, my remarks have been answered in a satisfying way. For these reasons I consider that this paper should be published in nature communication.

Version 1:

Reviewer comments:

Reviewer #1

(Remarks to the Author)

The authors have addressed my remaining comments to my satisfaction. I recommend publication of their manuscript in Nature Communications.

Reviewer #2

(Remarks to the Author)

The paper demonstrates proton flux enhancement with broad bandwidth and using traditional ion acceleration method. The plasma magnetic confinement of MeV proton beam may play an important role in beam collection, even it will not help to increase the total conversion efficiency from laser to the output ion beam.

My remarks have been answered in a satisfying way, I consider that this paper should be published in nature communication.

Response to referee comments:

We thank the referees for their review of our revised manuscript. We are pleased that the referees recognise the importance of the paper and improvement of the manuscript through their previous suggestions. We hope that with these answers to their queries and adjustments based on their comments, the manuscript will satisfy their expectations.

Our point-by-point response to all the referees' comments is given below:

Referee #1 (Remarks to the Author):

The authors have made significant improvements to their manuscript and have responded to my comments and questions adequately, which I appreciate. I have a few remaining comments and questions regarding their line of argument about co-propagating electrons.

We are pleased that the referee is satisfied by the improvements to the manuscript and thank them for their contribution to this. We address their questions regarding the co-propagating electrons below.

They heavily rely on the Cowan 2004 paper to justify using ~ 200 eV co-propagating electrons in their explanation of the proton focusing effect and as input to simulation. I want to point out that, firstly, Cowan explicitly states that electrons are co-moving "during much of the acceleration", not during the drift phase (which is what the authors model in their OSIRIS simulation). In fact, Cowan shows in simulations that removing the co-propagating electrons after 1 cm doesn't affect the proton beam emittance, i.e., they don't contribute to the observed beam laminarity beyond 1 cm (well into the "drift phase").

We do not heavily rely on the 200eV co-propagating electron beams for our interpretation of the proton focusing. The electron temperature has an effect on the proton beam propagation in vacuum, as the temperature controls the strength of the space charge field, but they play no significant role for a beam propagating through a background plasma due to the additional shielding that the cold background electrons provide. We have used a wide range of electron temperatures for our simulations (200 eV to 20 keV) and the focusing effect occurs regardless. For the work presented here, the role of the co-propagating electrons is to mitigate expansion of the proton beam in the vacuum case where it would otherwise be dominated by space-charge. Regarding the removal of electrons, note that in Cowen et al. they state "*A global increase of the ion beam envelope is expected for the non- neutral beam but no significant difference in the inferred emittance can be seen between the two shots.*" i.e., there is space-charge driven increase in *divergence* without compensation, consistent with our conclusions and simulations. That the emittance is (relatively) unaffected by the removal of electrons is not inconsistent with an increase in divergence provided the fields are not too nonlinear.

In order to clarify this point, we have slightly amended the description of the co-propagating electrons in the manuscript, as shown below (and highlighted in red in the revised document). "*The electrons were initialised with the same spatial distribution and forward drift velocity as the proton beam but with a thermal uncorrelated momentum spread corresponding to a temperature of 200 eV transverse to the propagation direction. This temperature is*

comparable with estimates of proton beam temperature inferred from measurements of beam emittance \cite{Cowan2004PRL} and is sufficiently low to prevent a space-charge driven increase in proton beam transverse momentum during vacuum propagation."

Second, Cowan describes 15-200 eV as the "proton transverse temperature", rather than the electron temperature, which is what the authors are using these numbers for. They should justify why they assume the electron temperature is the same as the proton transverse temperature.

A robust measurement of the characteristics of the co-propagating electron population is lacking in the literature. However, temperature of the rear surface plasma has been inferred from measurements of plasma expansion [Antici et al., PRL, 101, 105004 (2008)], indicating a hot (hundreds keV) and cold (tens of eV) electron population. While the hot electron temperature is typically used to determine the accelerating sheath fields, we have conservatively assumed that the electrons that are pulled along by the charge of the propagating proton beam must be colder. Hotter electron temperatures would, indeed, make the shielding effect from the background more pronounced. In the absence of more concrete measurements/theory we have therefore equated the temperatures of the protons and co-propagating electrons. While this is likely to represent a minimum electron temperature [Kemp et al., PRE, 75, 056401 (2007)] we emphasize that simulations have been performed with a variety of transverse electron temperatures for the co-propagating electron population, as mentioned in the previous response, with no significant differences in the case of the beam propagation through the vapour background (note that within the vacuum the higher transverse temperature of the electrons leads to strong sheath fields at the edge of the proton beam).

Third, as the authors point out, electrons would have to have keV energies to keep up with the protons - does that mean that in their simulations, protons outrun electrons fairly quickly and there is actually only a brief co-propagation phase? In that case they may need to revise their explanation of the effect, where the co-propagating electrons play a minor role (because co-propagation is short).

No, this is not correct. A *fluid* velocity corresponding to keV energy is required for the electrons to co-drift with the ions, but this is not the same as *temperature*, which represents the spread of velocities about the mean velocity. The electron temperature of 200 eV is defined in the rest frame of the fluid with both electron and ion fluids having the same mean velocity such that the net current is zero. Hence, the protons do not outrun the electrons.

As discussed the focusing mechanism is due to the formation of an azimuthal magnetic field around the propagating proton beam and is insensitive to the chosen electron temperature.

I am happy to recommend this manuscript for publication in Nature Communications if the authors can address these remaining comments.

We thank the reviewer again for recognising the suitability of this work for publication in Nature Communications.

Referee #2 (Remarks to the Author):

This paper demonstrates the generation of multi-MeV proton beams with an unprecedentedly low divergence of 1° (≤ 20 mrad) from a novel ambient-temperature liquid sheet with high shot-to-shot stability due to plasma collimation. The multi-Hz generation of laser-accelerated proton beams from laser to proton beam, it represents a step in advancing this source for future applications, as the proton flux or intensity is obviously enhanced by nearly 100 times compared to traditional TNSA regime. However, it shows approximately 0.5% of the 190 mJ laser pulse energy was converted to proton beam, it means the acceleration mechanism is still in traditional TNSA regime. Therefore I still have some concerns as following:

As we saw in Physics of Plasmas 2013 Vol. 20 Issue 7, there were experimental studies of divergence of proton beams from nanometer thick diamond-like carbon foils irradiated by an intense laser with high contrast. Proton beams with also small divergence of 2 degrees are observed in addition with a well-collimated feature over the whole energy range, this also constitutes the smallest value nearly the same as in this paper, as a consequence, ~ 100 times increase in proton flux was also observed, while even with higher conversion efficiency from laser to proton beam.

We are pleased that the referee appreciates the importance of the “unprecedentedly low divergence” and that “it represents a step in advancing this source for future applications”. As discussed in the paper and the previous round of referee responses, we believe that our experiment presents a significant step for the field in simultaneously demonstrating high flux, low divergence beams with high stability and kHz compatible systems. While the results presented in Bin et al., PoP, (2013) are very interesting and demonstrate low-divergence proton beams, as we comment in the manuscript, these measurements are made in a very different interaction regime. Crucially the results presented by Bin et al. required ultra-high laser contrast ($1E-9$ achieved using a double plasma mirror system) and ultra-thin target foils (<20 nm), for which there is no near-term solution for operating at >10 Hz repetition rates. Therefore, we maintain that our work opens up new possibilities for laser driven ion acceleration, despite still being in the TNSA regime. The observation and physical description of the collimation mechanism will also be of general interest. For these reasons we believe our results will not only inspire the plasma-accelerator community, but will be of significant interest to plasma physicists, including astrophysicists and fusion scientists, and radiation users (e.g. material scientists, radiobiologists), thereby warranting publication in Nature Communications.

Ionisation of the low-density vapour behind the target acts to limit the divergence growth of the proton beam, due to the protons propagation through the plasma where they are subject to magnetic pinching, this regime is maybe not applicable in higher laser intensity case and higher proton energy such as 60MeV, there are not any comments and extra simulations.

In the revised manuscript, we again focused on simulations to explain the observed experimental results. There is no reason to believe that the mechanism will not be applicable to higher energy proton beams, as the underlying concept is a response to a proton beam propagating through a plasma. Supporting this, below we present simulations for the propagation of a 60 MeV proton bunch through a background vapour. While the strength of

the generated magnetic field, the level of ionisation and the subsequent evolution of the proton beam will vary with energy, the magnetic focusing field is still formed and focuses the proton beam. As we point out in the paper further research is needed to determine how the focusing effect can be optimised for different interaction parameters, for example by tailoring the neutral vapour profile. We believe this is an exciting area for future study that is beyond the scope of this manuscript.

Figure: PIC simulations of 60 MeV proton beam propagating through a neutral vapour with constant vapour density illustrating formation of focusing magnetic fields and proton phase space modification leading to reduced transverse momentum.

We note that while acceleration to higher proton energies expands the applications that can benefit from these sources, the modest capabilities of the laser that was used during the presented experiment mean that such systems are far more accessible, both economically and in footprint, than the higher energy (multi-joule) lasers exploited to reach 60 MeV. Our novel approach can enable table-top few MeV proton sources at multi-Hz repetition rates for societal applications in materials analysis and nuclear medicine.

The paper demonstrates proton flux enhancement by plasma magnetic confinement of MeV proton beam, instead of new acceleration mechanism, which can also be realized by conventional magnets or capillary. I would recommend publication in other more technical journal.

Proton beam transport is a significant problem in the exploitation of laser-accelerated proton beams for applications and is not as trivial as the referee suggests. While conventional magnetic optic or active plasma lenses have been tested, these sit at some distance from the source with a limited acceptance aperture and are generally highly chromatic. This leads to poor efficiency in the capture of the proton beam and damage to the optics due to bombardment by x-rays and protons outside of the collection angle as well as taking up significant amounts of space and obstructing other diagnostics. The work presented in this manuscript represents an experimental demonstration that a so far unexplored self-driven focusing can be exploited to stably focus a TNSA proton beam with large energy spread over many centimetres of propagation. Not only this but our technique is compatible with high-

repetition rate targetry, allowing easy exploitation of newly available multi-Hz high power laser systems, while being minimally invasive and damage-resistant. We do not think that this is a purely technical result, given the novelty of the experimental results and the previously unexplored physical mechanism in high power laser experiments.

Referee #3 (Remarks to the Author):

The novel version of the manuscript by M. Streeter et al., considerably gained in clarity, both on the underlining physics and on the scope of the paper.

The focusing mechanism that is responsible, according to the authors' evidence and analysis, for the collimation of the proton beam certainly deserves to be disclosed to the laser-plasma accelerator community, in order to stimulate further exploration. As far as I'm concerned, my remarks have been answered in a satisfying way. For these reasons I consider that this paper should be published in nature communication.

We thank the reviewer for their contribution to the paper facilitated by their previous comments.

We would like to take the opportunity to thank all the reviewers again for their comments which have guided significant improvements to the manuscript.